# Effect of Al Content and Local Chemical Order on the Stacking Fault Energy in Ti–V–Zr–Nb–Al High-Entropy Alloys Based on First Principles

**DOI:** 10.3390/ma18092053

**Published:** 2025-04-30

**Authors:** Mengyao Chen, Xiaowen Yang, Xinpeng Zhao, Cheng Wen, Haiyou Huang

**Affiliations:** 1China Institute of Atomic Energy, Beijing 102413, China; chenmengyao1005@163.com; 2Institute for Advanced Materials and Technology, University of Science and Technology Beijing, Beijing 100083, China; yangxw200116@163.com (X.Y.);; 3College of Mechanical Engineering, Guangdong Ocean University, Zhanjiang 524000, China; 4Beijing Advanced Innovation Center for Materials Genome Engineering, University of Science and Technology Beijing, Beijing 100083, China

**Keywords:** lightweight high-entropy alloys, stacking fault energy, density functional theory, short-range order

## Abstract

As a promising candidate for next-generation aviation structural materials, lightweight refractory high entropy alloys (HEAs) exhibit high strength, low density, and excellent high-temperature performance. In this study, we investigated the influence of local chemical ordering on the properties of Ti–V–Zr–Nb–Al HEAs using Monte Carlo (MC) simulations based on density functional theory (DFT) calculations. We established that the chemical short-range ordering (SRO) in Ti–V–Zr–Nb–Al HEAs increases with the Al content, resulting in a gradual increase in stacking fault energy (SFE). This theoretical investigation suggests that SRO can be utilized to tailor the performance of HEAs, thereby providing guidance for the scientific design of macroscopic mechanical properties.

## 1. Introduction

High-entropy alloy (HEA) is a new type of material, initially proposed by Yeh [1] and Cantor [2] in 2004. It is composed of more than five elements, with each atomic fraction between 5% and 35% and randomly occupying one crystallographic site. Refractory high-entropy alloys (RHEAs) were first proposed in 2010 with the primary purpose of withstanding high temperatures as an alternative to nickel-based high-temperature alloys and for other applications such as in the manufacturing of gas turbine blades, armor, and aerospace materials [3]. Because RHEAs exhibit many ideal performance traits that traditional alloys cannot achieve, they have attracted growing interest [4,5,6]. RHEAs contain mainly refractory elements such as W, Ta, Mo, Zr, Hf, V, and Nb, as well as some low melting point elements such as Al and Cr [7].

Among the various RHEAs, Ti–V–Zr–Nb–Al alloy shows considerable potential due to its excellent properties such as light weight and high strength, endowing it with broad application prospects for use in nuclear reactors and in aerospace fields. Previously reports regarding Ti–V–Zr–Nb–Al HEAs indicate that such alloys usually exhibit high strength, but their structural plasticity is often not prominent [8]. For these types of alloys, typical long planar slip bands are often formed, and dislocations usually accumulate in these narrow planar slip bands, resulting in severe deformation localization [9]. This is related to the performance of dislocations in BCC. The dislocation motion is controlled by the stacking fault energy (SFE) [10], which is an effective parameter for quantitatively characterizing the difficulty of the dislocation occurrence, exerting an essential effect on the defects, defect clusters, and the mobility of the dislocations. Generally speaking, the BCC structure of pure metals such as Mo, W, and V displays a lower SFE, enabling activation of cross-slips and multiple-slip systems, which indicates that the interactions of dislocations are more complicated [11]. Furthermore, the dislocation decomposition is relatively tricky, and it is easy to form stable pinned structures such as trilobal dislocations, which are unfavorable to ductility [12]. In previous studies, the excellent mechanical properties of Ti–V–Zr–Nb–Al HEA have been attributed to the local geometrical coordination effect of deformation, resulting in a different nature of SFE from that of conventional BCC metals [13,14,15].

It has been shown that better performance of BCC-structured HEA may be related to local chemical fluctuations. The structure of Ti–Zr–Nb–Al alloys is strongly controlled by the Al content. When the Al content is low, Al takes the place of other elements on the BCC matrix and maintains the BCC structure; when the Al content is high, it tends to form Al–Zr ordered phases and may further lead to phase segregation in the BCC matrix [16]. Therefore, the Al content may lead to the formation of critical localized chemical ordering in the lattice, profoundly affecting the magnitude of the SFE. However, experiments can only measure the magnitude of intrinsic stacking fault energy. In contrast, generalized stacking fault energy (GSFE) simultaneously controls the deformation-coordinated competition of different defects and thus indicates a deeper mechanism of mechanical modulation [7,10,17]. Unfortunately, the unstable layer fault energy items included in the GSFE can only be obtained by numerical calculation methods, so a precise theoretical study related to the mechanism of localized chemical ordering formation and the regulation mechanism of SFE in Ti–V–Zr–Nb–Al HEA is still missing.

In this work, we perform detailed first-principles calculations to elucidate the effect of Al content and local chemical order on the SFE of Ti–V–Zr–Nb–Al HEA, with a focus on understanding how chemical short-range order (SRO) affects the variation in SFE in these high-entropy alloys based on different Al contents and offer valuable guidance for the design and development of next-generation high-performance materials.

## 2. Methodology

### 2.1. DFT Calculations

In the present work, density functional theory (DFT) [18] calculations were executed utilizing the Vienna ab initio Simulation Package (VASP 5.4.4) [19,20]. The interaction between core electrons was represented through the projector augmented wave (PAW) method [21]. The exchange-correlation function was approximated using generalized-gradient approximation (GGA) via Pardew–Burke–Ernzerhof (PBE) [22] parameterization. A plane wave basis set cutoff energy of 300 eV was applied. For the Monte Carlo (MC) simulations, a 1 × 1 × 1 k-point mesh was used [23], whereas for SFE calculations, a 4 × 4 × 1 mesh was employed. The semi-core p electrons of all elements were accounted for as valence electrons, where applicable [21,24].

### 2.2. DFT-Based Monte Carlo Simulations

To design structures that represent HEA, MC simulations were utilized. These simulations incorporated swap trials for each atom, with acceptance probabilities based on the Metropolis–Hastings sampling [25] method. Supercells were meticulously designed, each containing 80 atoms for the {110} and {112} planes and 56 atoms for the {123} planes. These specially crafted quasi-random structures were employed as the initial configurations for simulations, with the simulation temperature set at precisely 300 K. Energy calculations for each structure were performed using the VASP. The MC simulations were executed for a total of 3000 steps for each structure, which equates to 38 to 53 swap trials per atom.

### 2.3. Local Chemical Parameter

The Warren–Cowley parameter (WCP) [26] was utilized to measure the degree of chemical order surrounding an individual atomic species. The calculation of the WCP was based on the following formula:(1)$${WCP}_{ij}=1-Z_{ij}/c_{j}Z_{i}$$
where Z_ij_ represents the count of j-type atoms surrounding i-type atoms, Z_i_ is the total count of atoms around i-type atoms, and c_j_ signifies the atomic fraction of j-type atoms in the HEA. A WCP value of zero indicates a perfectly random distribution. A positive WCP suggests a preference for reducing the number of i-j atomic pairs, whereas a negative value indicates an increase. In this study, the WCP was determined by tallying the types of nearest neighbor elements.

### 2.4. Stacking Fault Energy Calculations

These supercell models consisting of 20, 20, and 14 atomic layers were designed to investigate the unstable stacking fault energies along three specific planes: {110}, {112}, and {123}. The slip direction for all models is uniformly set along the <111> direction. These three slip systems will be referred to as I, II, and III, respectively. Each layer contains four atoms. The atomic positions were permitted to relax exclusively along the direction that is perpendicular to the slipping planes. In terms of boundary conditions, an infinitely large structure is ideal for the calculations. However, it is a common practice in the first-principles calculation packages to repeat a finite size supercell infinitely with periodic boundary conditions (PBC) [27]. A schematic explanation is depicted in Figure 1. The Monte Carlo simulation was performed on the volume (i.e., the entire spatial domain) rather than on separate planes. The displacement of the 20-layer supercell shown in Figure 1 is used for GSFE calculations to elucidate the formation of stacking faults. The SFEs of the Ti–V–Zr–Nb–Al HEA for the above-mentioned three typical slip systems are calculated as follows [28]:(2)$$\gamma_{SFE}=\frac{E_{f}-E_{p}}{S}$$
where E_f_ represents the total energy of the stacked configuration that incorporates the fault vector u. E_p_ denotes the total energy of the stacked configuration in the absence of slipping. S signifies the area within the supercell that is affected by the fault.

To further elucidate the collective influence of Al content and local chemical order on SFEs of Ti–V–Zr–Nb–Al HEA, we have calculated the average SFE using the following equation [29]:(3)$${\bar{\gamma }}_{SFE}=\sum_{j=1}^{3}\frac{{\left(\gamma_{SFE}^{j}\right)}^{2}}{\sum \gamma_{SFE}^{j}}$$
where j represents the three distinct slip systems under consideration.

## 3. Results and Discussion

### 3.1. Local Chemical Order in Ti–V–Zr–Nb–Al High-Entropy Alloys

To investigate the effect of local chemical ordering on the SFE, it is necessary to build a realistic chemical SRO model of the Ti–V–Zr–Nb–Al HEA. In an earlier computational work [30], simulations using the DFT/MC method established chemical SRO results that showed agreement with experimental measurements. Therefore, in this work, we use the DFT/MC method to develop numerical models of Ti–V–Zr–Nb–Al high-entropy alloys with varying degrees of SRO. Specifically, we used DFT/MC simulations to investigate the SRO of the {110}, {112}, and {123} planes in BCC-structured Ti–V–Zr–Nb–Al HEA, as described in the Methods section. The most significant trends in the change of potential using DFT/MC are shown in Figure 2a The potential energy curves seem to converge, although the number of exchange trials with each atom is relatively small compared to those in the classical MC simulation.

SRO has a significant effect on the microstructure of the alloy. We use WCP to characterize the trend of local chemical ordering obtained in the MC simulations. It is unfavorable for atom pairs when the value of WCP is positive, while it is favorable for atom pairs when the value of WCP is negative. The results indicate that different elements exhibit varying degrees of the segregation effect, which in some elements, is significantly more potent than in others. This tendency is captured by the WCP in Figure 2b–d, where some elements tend to form clusters, while others tend to have different neighbors.

For the Ti–V–Zr–Nb–Al HEA with different Al contents along the {110} plane, there is a strong tendency to form Al–Al pairs (WCP < 0). This tendency decreases with increasing Al content, but remains relatively significant (WCP < −1). For other elements, Nb–X pairs are more unfavorable (X = Ti, V, Al), with WCP > 0. A similar pattern is observed for the {112} plane in these alloys, as shown in Figure 2c, with a pronounced tendency to form Al–Al pairs (WCP < 0). Zr–Al, Zr–V, and Zr–Nb pairs also exhibit strong formation trends, whereas V–Al, Nb–Al, and Ti–Nb pairs are unfavorable (WCP > 0). As shown in Figure 2d, the Al–Al, Zr–Nb, and Ti–V pairs show a significant tendency to form pairs for the {123} plane, while other pairs occur in a random distribution, as most of their WCPs are near 0.

### 3.2. Stacking Fault Energy of Random Solid-Solution Ti–V–Zr–Nb–Al Alloys

Stacking fault is equivalent to a movement of a crystal layer with a specific distance relative to its immediate neighbors, usually located along the closest-packed direction within the closest-packed plane. The atomic occupancy in the close-packed plane ideally does not change, so the most significant impact on the lattice must be the direction perpendicular to the stacking fault [31]. Based on this regulation and the change in system energy, we obtain the curve of GSFE versus the displacement of the upper parts. To demonstrate the visualization mechanism of the effect of Al content on the SFEs in Ti–V–Zr–Nb–Al HEA, we investigated the distribution of the unstable SFE in high-entropy alloys with different Al contents along three specific planes: {110}, {112}, and {123}.

Figure 3a–c show the SFE curves calculated by the first principles of Ti–V–Zr–Nb–Al high-entropy alloys for the three typical slip systems mentioned previously. The maximum energy on the SFE curve is the unstable SFE, which determines the energy barrier for the nucleation of trailing dislocations [32]. It occurs at 0.5b^<111>^ for the {110}, {112}, {123} planes with different Al contents for Ti–V–Zr–Nb–Al HEA. Therefore, we chose the SFE at 0.5b^<111>^ for the following research.

According to previous literature [33], it is known that the main slip planes are the {110} and {112} planes in most of the body-centered cubic crystals, and the slip plane of {123} is also commonly observed in practical experiments. The actuation order of the slip system is {110}, {112}, then {123}. As shown in Figure 3d, we analyzed the SFEs of the {110}, {112}, and {123} planes in Ti–V–Zr–Nb–Al high-entropy alloys with different Al contents. For {110} and {112}, the SFEs of the alloys gradually increase with the increase in Al content, which indicates that dislocations are more likely to slip along {110} rather than along {112}. For the {123} planes, Figure 3d displays that the SFEs decreased with the addition of Al (e.g., 0–14%), and the slip will occur in preference to that of the other two modes. When the Al content was 0%, the SFEs of the {110}, {112}, and {123} planes in Ti–V–Zr–Nb–Al high-entropy alloys increased gradually. For Al = 5%, the SFE of the {112} plane is larger than in the {110} plane. However, when Al = 7%, the SFE of the {123} plane is much smaller than those of the {110} and {112} plane. Similarly, when Al = 10%, the SFE of the {112} plane is larger than that of the {110} plane, but when Al = 14%, the SFE of the {123} plane is much smaller than that of the {110} and {112} plane. This indicates that preferential initiation of slip occurs in the {123} planes. According to the research in Ref. [17], the magnitude of the stacking fault energies (SFEs) affects the deformation mechanism of the materials, where the larger the value of the SFEs, the more likely the cross-slip of dislocations will occur during the deformation process, and conversely, the lower the SFEs, the more prone the dislocations are to planar slip. Vipul [34] investigated the influence of the Al content on the mechanical properties of Ti–Zr–Nb–Hf refractory high-entropy alloys. As the Al content increased from Al-0 to Al-1, the yield strength increased from 310 Mpa to 1245 Mpa. Additionally, the hardness data of the alloys exhibits a similar increasing trend. The experimental results effectively demonstrate the influence of the SFEs on the properties of refractory high-entropy alloys. Therefore, the SFEs can be widely used to predict or simulate the mechanical properties of BCC high-entropy alloys and to guide the development of high-performance materials.

### 3.3. Local Chemical Order Influence on Stacking Fault Energy

We calculated the variation in SFE of the {110}, {112}, and {123} planes in Ti–V–Zr–Nb–Al HEA with different Al contents as a function of the MC simulation steps. As shown in Figure 4a–c, the SFE curves tend to flatten when the MC simulation steps reach 2000–3000, indicating that the system structure reaches equilibrium. Based on this equilibrium state, we calculated the energy obtained from the structural optimization of SRO stacking structures for Ti–V–Zr–Nb–Al HEA with different Al contents, as shown in Figure 4a–c.

The SFE of the {110}, {112}, and {123} planes of Ti–V–Zr–Nb–Al HEA with different Al contents can be quantitatively correlated with the degree of chemical SRO, reflected by the total non-proportional number of local atomic pairs, WCPsum, as shown in Figure 5a–c, respectively. For most HEAs, the more significant the WCPsum, the more ordered the samples, and the higher the SFE; that is, the SFE increases gradually with the increase in the SRO of the HEA. Recent work [13,35] has proven that the presence of SRO affects the periodic Peierls–Nabarro energy barriers and SFE distribution, thus giving rise to considerable fluctuated lattice resistance, complex dislocation maneuvers, and TWIP/TRIP behaviors in HEAs. This finding appears to imply that the experimentally studied SRO affects the variation in the SFE of the HEA, which falls within the range investigated by our DFT calculation, and researchers can adjust the performance change in the HEA by adjusting the change in the SRO.

## 4. Conclusions

This work combines DFT and MC simulations to investigate the effects of Al content and local chemical ordering on the SFE in Ti–V–Zr–Nb–Al refractory high-entropy alloys. The key findings are summarized as follows:

MC simulations reveal significant SRO in the {110}, {112}, and {123} planes of the alloys, particularly around the Al atoms. While increasing Al content reduces the tendency for Al–Al pair formation, strong Al–Al pair clustering persists. In BCC crystals, the primary slip systems typically activate in the order of {110}, {112}, and {123}. For Ti–V–Zr–Nb–Al HEA, the SFE of the {110} and {112} planes increase with higher Al content, favoring slip initiation on the {110} plane. In contrast, the {123} plane exhibits a decreasing SFE trend with Al addition, suggesting preferential slip activity compared to that of the other planes. Additionally, the system reaches structural equilibrium when the MC simulation steps exceed 2000–3000, as evidenced by the stabilization of the SFE curves. Furthermore, chemical SRO significantly modulates the SFE values, highlighting the critical role of local atomic configurations.

Overall, Al content variation alters the local chemical order in Ti–V–Zr–Nb–Al HEA, which not only governs SFE, but may also influence other defect-related properties (e.g., vacancy behavior). These findings suggest that tailoring elemental compositions to control chemical SRO could serve as an effective strategy for optimizing the mechanical performance of refractory HEAs, thereby guiding future alloy design.

## Figures and Tables

**Figure 1 materials-18-02053-f001:**
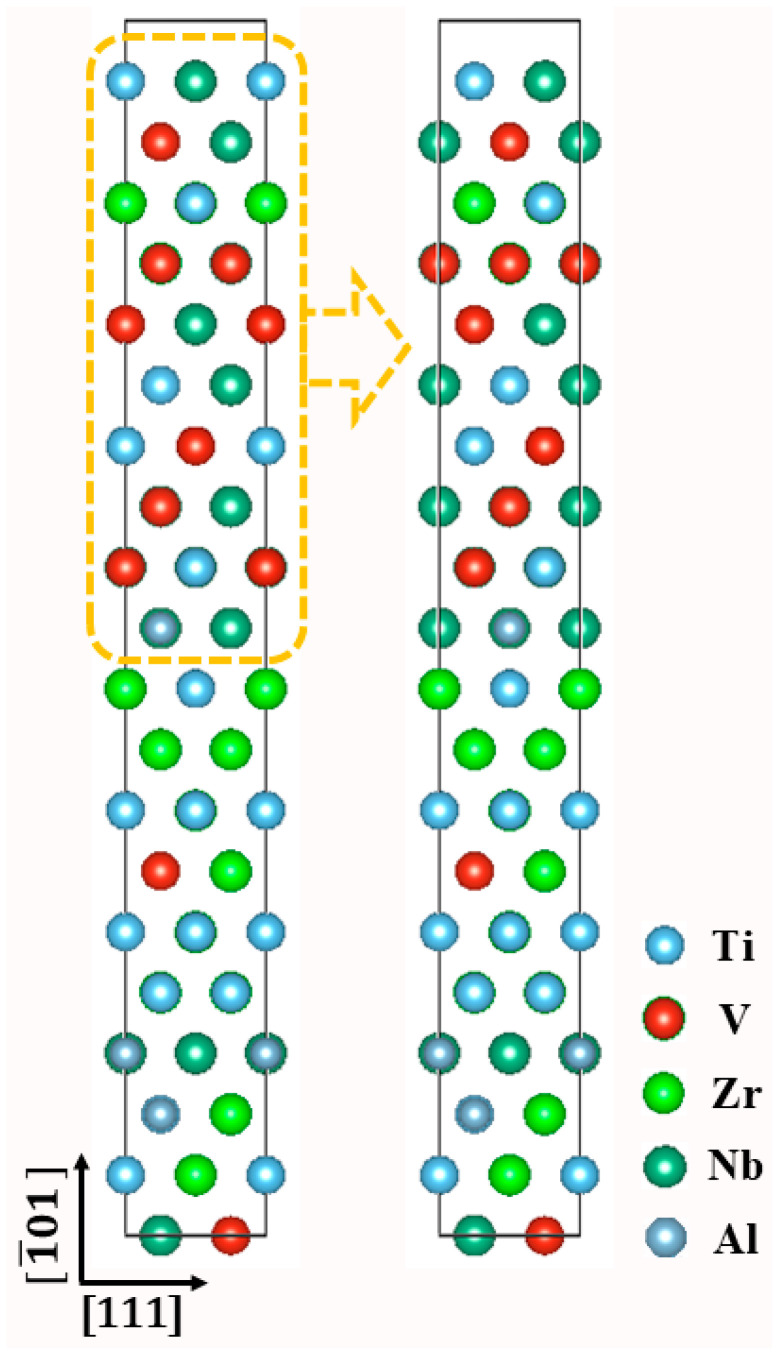
Displacement of the slip system to explain the formation of stacking fault.

**Figure 2 materials-18-02053-f002:**
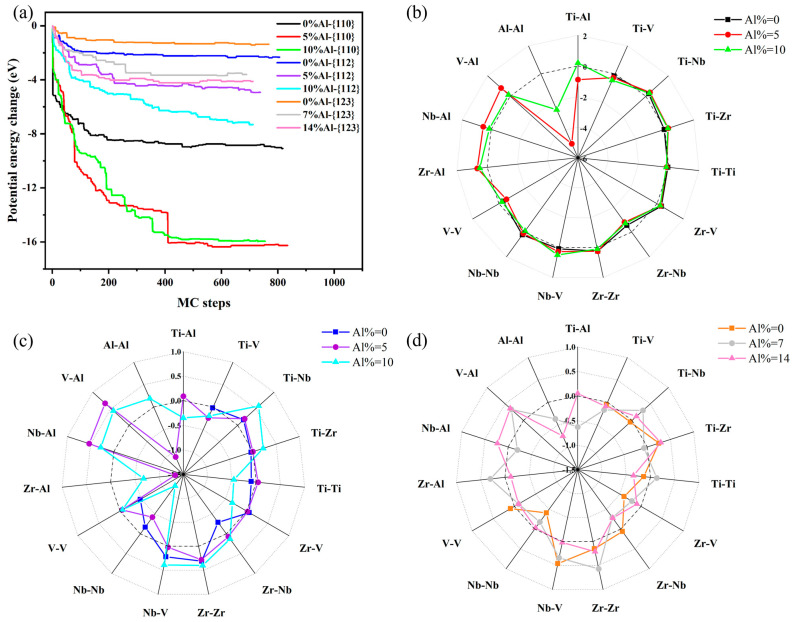
Evolution of energy and local chemical ordering in Ti–V–Zr–Nb–Al HEA: (**a**) potential energy changes with the steps of the Monte Carlo simulation (MC steps); detailed values of Warren–Cowley parameter (WCP) in the nearest neighbor shell for all atom pairs in the (**b**) {110} planes, (**c**) {112} planes, and (**d**) {123} planes of Ti–V–Zr–Nb–Al HEA with different Al contents, where the lines and balls represent HEAs with short-range order (SRO), and the dashed lines represent an ideal random solid solution, while WCP is not defined when Al = 0%.

**Figure 3 materials-18-02053-f003:**
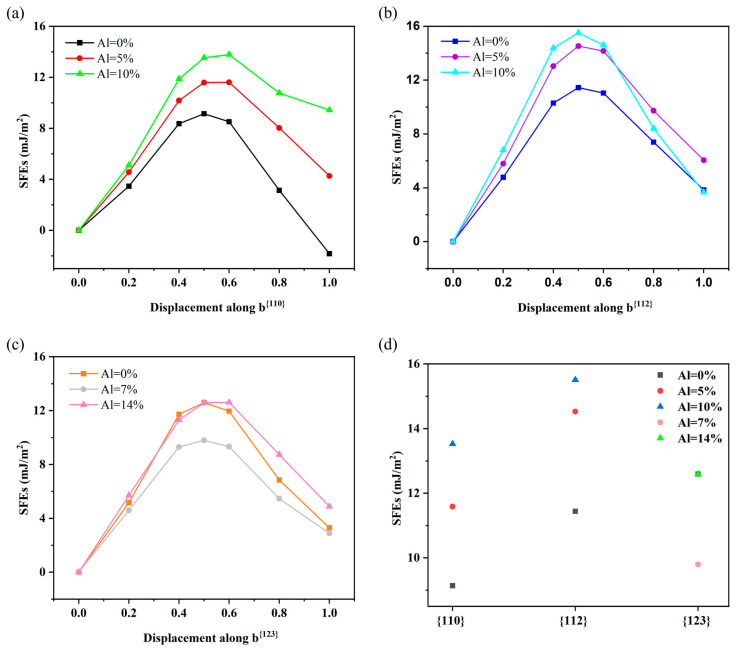
The first-principles calculated generalized stacking fault energy (GSFE) curves in the (**a**) {110} plane, (**b**) {112} plane, and (**c**) {123} plane of Ti–V–Zr–Nb–Al HEA with different Al contents. (**d**) The GSFE of Ti–V–Zr–Nb–Al HEA with different Al contents based on different planes. The slip direction for all models is uniformly set along the <111> direction.

**Figure 4 materials-18-02053-f004:**
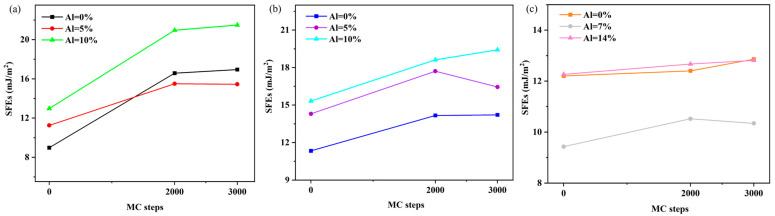
Evolution of stacking fault energy of (**a**) {110}, (**b**) {112}, and (**c**) {123} planes in Ti–V–Zr–Nb–Al high-entropy alloys with different Al contents as a function of Monte Carlo simulation steps (MC steps).

**Figure 5 materials-18-02053-f005:**
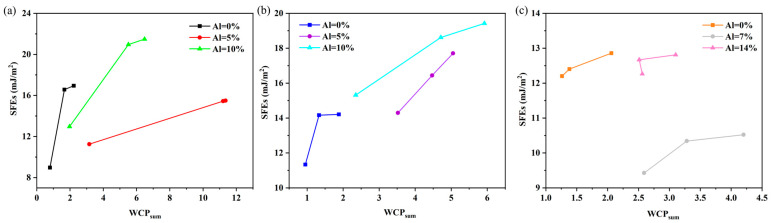
SFE of (**a**) {110}, (**b**) {112}, and (**c**) {123} planes in Ti–V–Zr–Nb–Al HEA with different Al contents were plotted versus the results for WCPsum.

## Data Availability

The raw data supporting the conclusions of this article will be made available by the authors on request.

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
