# Peer review of "Effect of Al Content and Local Chemical Order on the Stacking Fault Energy in Ti–V–Zr–Nb–Al High-Entropy Alloys Based on First Principles"

_materials, 2025, doi:10.3390/ma18092053_

Round 1

Reviewer 1 Report

Comments and Suggestions for Authors

Comments on the Quality of English Language

It can be improved by writing short sentences.

Reviewer 2 Report

Comments and Suggestions for Authors

In the manuscript «Effect of Al content and local chemical order on the stacking fault energy in Ti-V-Zr-Nb-Al high-entropy alloys» investigated the properties of the local chemical order through Monte Carlo (MC) simulations based on density functional theory (DFT) calculations, and established the influence of Al content on the stacking fault energy (SFE) of Ti-V-Zr-Nb-Al HEA, focusing on how the chemical short-range order (SRO) affects the variation of the SFE. The manuscript presents interesting results that deserve publication, but there are several comments:
1. References 3, 7, 10, 16 have low relevance. Reference 3 does not consider high-temperature properties, reference 8 has no information on slip systems, reference 10 has no information on dislocation slip, and reference 16 considers fcc materials. References 7 and 17 are repeated.
2. It is not clear from part 2 whether the study of the local chemical environment was carried out on planes separately or in volume, and the planes are presented for illustration? What were the boundary conditions in the case of planes? Additions are required in part 2.
3. In Figure 2, at a concentration of Al=0, the parameter WCP is not defined (0/0), but is marked as zero on the graphs. Within what limits can this parameter change in the studied compositions? In Figures 2c and 2d, there is no series of values (the curve is broken), why? Explanations are needed.
4. Line 188 needs to clarify aluminum concentrations.
5. After Figure 4 comes Figure 3, which must be brought into line.

Comments on the Quality of English Language

Minor Editing of English language required.

Reviewer 3 Report

Comments and Suggestions for Authors

Dear Authors, 

Please find below comments on the paper “Effect of Al content and local chemical order on the stacking fault energy in Ti-V-Zr-Nb-Al high-entropy alloys”. This article describes the influence of aluminum content on the Stacking Fault Energy of Ti-V-Zr-Nb-Al HEAs using Monte Carlo (MC) simulations based on density functional theory (DFT) calculations. While the investigation is based solely on computational methodologies, the manuscript aligns with the scope of Materials. The work would interest our readers due to the ongoing research on designing high-entropy alloys. The manuscript presents results of good quality in the field. However, editing is needed to enhance the clarity, rigor, and contextualization of their findings.

Please follow the suggestions:

Given the absence of experimental data validation, the title should explicitly indicate the computational nature of the study. Thus, please improve the title “Effect of Al content and local chemical order on the stacking fault energy in Ti-V-Zr-Nb-Al high-entropy alloys” to include terminology related to the DFT computational approach.

The authors are encouraged to compare their computational results more comprehensively with existing literature, encompassing both experimental and theoretical investigations of staking fault energies and related properties in similar constituent alloy systems. The discussion should move beyond a mere citation of references and explicitly highlight the novel contributions and unique insights offered by this study in the current state of knowledge.

The authors must address these suggestions thoroughly before the manuscript can be considered for publication in Materials.

Author Response

Comments 1. Given the absence of experimental data validation, the title should explicitly indicate the computational nature of the study. Thus, please improve the title “Effect of Al content and local chemical order on the stacking fault energy in Ti-V-Zr-Nb-Al high-entropy alloys” to include terminology related to the DFT computational approach.

Response 1: Thank you for your comment. The title has been revised and additional terms related to DFT calculation method have been added, specifically “Effect of Al content and local chemical order on the stacking fault energy in Ti-V-Zr-Nb-Al high-entropy alloys based on First-principles”

Comments 2. The authors are encouraged to compare their computational results more comprehensively with existing literature, encompassing both experimental and theoretical investigations of staking fault energies and related properties in similar constituent alloy systems. The discussion should move beyond a mere citation of references and explicitly highlight the novel contributions and unique insights offered by this study in the current state of knowledge.

Response 1: Thank you for your comment. We are extremely grateful for your taking the time to carefully review our paper. Your valuable comments are of great significance for the improvement of the paper. We have conducted in-depth thinking and supplementary work based on your suggestions.

We have comprehensively reviewed the experimental research literature on the stacking fault energy and related properties in the systems of similar component alloys in Page 6, line 206. We have added the citation of Reference 34 and made a detailed comparison between our computational results and the experimental data presented in this reference.

According to the research[17], the magnitude of the Stacking Fault Energies (SFEs) affects the deformation mechanism of materials, where the larger the value of the SFEs, the more likely the cross-slip of dislocations will occur during the deformation process, and conversely, the lower the SFEs, the more prone the dislocations are to planar slip. Vipul[34] investigated the influence of the Al content on the mechanical properties of TiZrNbHf refractory high-entropy alloys. As the Al content increased from Al-0 to Al-1, the yield strength increased from 310 Mpa to 1245 Mpa. Additionally, the hardness data of the alloys exhibits a similar increasing trend. The experimental results effectively demonstrate the influence of the SFEs on the properties of refractory high-entropy alloys. Therefore, the SFEs can be widely used to predict and simulate the mechanical properties of BCC high-entropy alloys and to guide the development of high-performance materials.

The revised content is highlighted in the paper using in red font for your convenient review. Thank you again for your valuable comments. We hope the revised paper meets the publication requirements.

References

[17] V. Vítek. Intrinsic stacking faults in body-centred cubic crystals. Philos Mag. 18, 773-786(1968).

[34] Vipul Bhardwaj, Qing Zhou,Fan Zhang, et al. Effect of Al addition on the microstructure, mechanical and wear properties of TiZrNbHf refractory high entropy alloys. Tribol. Int. 160, 107031.1-11 (2021).

Round 2

Reviewer 1 Report

Comments and Suggestions for Authors

The reviewer's comments were responded nicely.

Author Response

No further comments need to reply.

Reviewer 2 Report

Comments and Suggestions for Authors

The comments have been corrected and the manuscript can be accepted.

Comments on the Quality of English Language

Minor editing of English language required.

Author Response

No further comments from reviewer need to reply